# The Effect of Firewood Moisture Content on the Atmospheric Thermal Load by Flue Gases Emitted by a Boiler

**Ladislav Dzurenda and Adrian Banski \***

Faculty of Wood Sciences and Technology, Technical University in Zvolen, 96001 Zvolen, Slovakia; dzurenda@tuzvo.sk

\* Correspondence: banski@tuzvo.sk; Tel.: +421-45-5206-368

**Abstract:** In this paper, we present an analysis on the effect of the moisture content of firewood on the atmospheric thermal load created by the heating of flue gases with temperatures of $t_{fg}$ = 120–200 °C, emitted by a boiler when wood with moisture content of W = 10%–60% is combusted. The load of the atmosphere created by the heat of the flue gases with temperatures of $t_s$ = 120 °C from the boiler, where dried wood with the moisture content of W = 10% is combusted, is Q = 9.2 MJ·GJ$^{-1}$. The atmospheric thermal load caused by flue gases with the temperature of $t_s$ = 200 °C, resulting from the combustion process of wet firewood with a moisture content of W = 60%, is 3.8 times higher compared with the above-mentioned conditions. The heating of water vapor from the evaporated water occurring in combusted wood, as well as the heat of the heated nitrogen and unoxidized oxygen in the combustion air delivered to the furnace of a firewood boiler, are considered to be reasons for the increasing atmospheric thermal load caused by the heating of flue gases, resulting from the combustion of wood with higher moisture content.

**Keywords:** environment; firewood; moisture content; boiler; heat of combustion products; atmosphere

## 1. Introduction

The wood of deciduous trees, with its energetic properties in the seasoned state, and which is in accordance with the standard of *STN EN 14961 Solid biofuels*, is a biofuel with a heat value of $Qn$ = 18.1 MJ·kg$^{-1}$, a high percentage of volatile flammable substance V = 85%, and a low ash content of A = 0.3%. In comparison to fossil fuels, the ash content of firewood from energy plantations or from forests is 15–30 times lower than the ash content of coal [1–5]. This can be considered a positive energetic property of firewood. The affinity to water and water vapor is a negative feature of firewood. The relative moisture contents of a freshly felled tree in the dormant season ranged from W = 35%–65%, depending on the wood species [1,6–8]. Firewood in branch form stacked under cover or against a sheltering wall dries naturally to an air-dried state, i.e., the moisture content of W = 18%–25% [6,9]. Unprocessed wood waste with a moisture content of W = 10%, or biofuel in the form of briquettes and pellets [10–12] made from wood with a specific size and moisture content, play an important role in terms of energy efficiency.

According to the authors [13–17], the efficiency of producing heat from firewood depends on the construction of the heat generator as well as on the energetic properties of firewood and energetic and environmental benefits delivered by the boiler. The energetic properties of firewood depend especially on its moisture content. The basic energetic properties include the gross calorific value ($Qs$) and heat value ($Qn$), but also the burning process in the furnace, including the flame temperature, amount of flue gases created, dew-point temperature of flue gases, and emission production, which are all

affected by the wood moisture content in a negative way. The construction of a boiler heat exchanger affects the use of the calorific value of flue gases, namely the cooling rate of the flue gases before they are delivered into the atmosphere. Currently, the energy efficiency of mid-efficient energy firewood boilers is $\eta_k$ = 80%–85%. On the other hand, the energy efficiency of modern biofuel boilers with guaranteed energetic properties is $\eta_k$ = 92%.

Recently, biofuels have been at the center of our attention, not only because of their market accessibility, energy, and economic efficiency, but also because of their environmental benefits, as the authors of [18–26] mention. The results of the research into the effect of moisture content of firewood, and the temperature of flue gases on the atmospheric thermal load by flue gases emitted by a firewood boiler are presented in the paper.

## 2. Model Evaluation of the Atmospheric Thermal Load by the Heat of Flue Gases

The heat present in the exhaust gases from the boiler to the atmosphere is the thermal load of the atmosphere. The heat of emitted the flue gas related to the production of 1 GJ of heat is mathematically described by Equation (1), as follows:

$$Q_{fg} = m \cdot V_{fg} \cdot c_{fg} \cdot \left( t_{fg} - t_{fg-e} \right) \qquad \left[ \text{MJ} \cdot \text{GJ}^{-1} \right] \tag{1}$$

The algorithm used to calculate the individual parameters of the atmospheric thermal load created by the flue gases is dependent on the chemical composition of the flammable substances of wood ($C^{daf}$, $H^{daf}$, $O^{daf}$, and $N^{daf}$), ash content in wood ($A$), relative moisture content of combusted firewood ($W$), excess of combustion air ($\lambda$), temperature of the flue gases delivered to the air from a boiler ($t_{fg}$), and temperature of the flue gases cooled to the temperature of the atmospheric air delivered to a boiler furnace ($t_{fg-e}$), which are described using the following equations:

The quantity of firewood burnt in the boiler furnace to produce the heat of 1 GJ is calculated using the following equation:

$$m_{p-1GJ} = \frac{10^6}{Q_n \cdot \eta_K} \qquad \left[ \text{kg} \cdot \text{GJ}^{-1} \right] \tag{2}$$

The specific volume of humid flue gases produced in the combustion process of 1 kg of firewood, depending on the mentioned parameters, is described using the following equation:

$$V_{fg} = \left[ 1.867 \cdot \frac{C^{daf}}{100} + 11.2 \cdot H^{daf} + 0.8 \cdot \frac{N^{daf}}{100} + V_{air} \cdot (\lambda - 0.21) \right] \cdot \left[ 1 - \frac{A}{100} - \frac{W}{100} \right] +$$
$$1.24 \cdot \frac{W}{100} \quad \left[ \text{m}_\text{n}^3 \cdot \text{kg}^{-1} \right] \tag{3}$$

The production of humid flue gases produced in the combustion process of firewood to produce the heat of 1 GJ is calculated using the following equation:

$$V_{fg-1GJ} = m \cdot V_{fg} \qquad \left[ \text{m}_\text{n}^3 \cdot \text{GJ}^{-1} \right] \tag{4}$$

The mean value of the specific heat capacity of 1 $\text{m}_\text{n}^3$ of flue gases when the pressure is constant is described by the following equation:

$$c_{fg} = c_{CO2} \cdot X_{CO2} + c_{CO} \cdot X_{CO} + c_{O2} \cdot X_{O2} + c_{N2} \cdot X_{N2} + c_{H2O} \cdot c_{H2O} \cdot X_{H2O} \qquad \left[ \text{kJ} \cdot \text{mn}^{-3} \cdot \text{K}^{-1} \right] \tag{5}$$

The values of the mean specific heat capacity of 1 $\text{m}_\text{n}^3$ of the *i*-th component of the flue gases when the pressure is constant ($c_{p-i}$) [ kJ.$\text{m}_\text{n}^{-3} \cdot \text{K}^{-1}$] are in Table 1. The values for $X_i$ are the volumetric proportions of the *i*-th component of the flue gases [-].

**Table 1.** Values of the mean specific heat capacity of 1 $m_n^3$ of flue gases when the pressure is constant. [14].

| Temperature [°C] | $c_{p\text{-}i}$–Mean Specific Heat Capacity of 1 $m_n^3$ of Flue Gases When the Pressure is Constant [kJ·$m_n^{-3}$·$K^{-1}$] | | | |
|:---:|:---:|:---:|:---:|:---:|
| | $CO_2$ | $O_2$ | $N_2$ | $H_2O$ |
| 0 | 1.620 | 1.306 | 1.302 | 1.491 |
| 100 | 1.725 | 1.319 | 1.306 | 1.499 |
| 200 | 1.817 | 1.336 | 1.310 | 1.520 |
| 300 | 1.892 | 1.357 | 1.315 | 1.537 |
| 400 | 1.955 | 1.382 | 1.327 | 1.557 |
| 500 | 2.022 | 1.403 | 1.336 | 1.583 |
| 600 | 2.077 | 1.419 | 1.348 | 1.608 |

The functional dependences of the mean specific heat capacity of 1 $m_n^3$ of the individual components of the flue gases on the temperature ($t$) when the pressure is constant are described using the following equations:

$$\text{Carbon dioxide } CO_2: C_{CO2} = 0.0008 \cdot t + 1.6473 \quad [kJ.m_n^{-3}.K^{-1}] \tag{6}$$

$$\text{Water vapor } H_2O: c_{H2O} = 10^{-7} \cdot t^2 + 10^{-4} \cdot t + 1.4895 \quad [kJ.m_n^{-3}.K^{-1}] \tag{7}$$

$$\text{Oxygen } O_2: c_{O2} = 5.10^{-8} \cdot t^2 + 2.10^{-4} \cdot t + 1.3036 \quad [kJ.m_n^{-3}.K^{-1}] \tag{8}$$

$$\text{Nitrogen } N_2: c_{N2} = 9.10^{-8} \cdot t^2 + 2.10^{-5} \cdot t + 1.3022 \quad [kJ.m_n^{-3}.K^{-1}] \tag{9}$$

The volumetric proportions of the individual components in the flue gases from the firewood combustion are determined using the following equations:

Volumetric proportion of carbon dioxide in flue gases

$$X_{CO2} = \frac{1.867 \cdot \frac{C^{daf}}{100} \cdot \left[1 - \frac{A}{100} \cdot \left(1 - \frac{W}{100}\right) - \frac{W}{100}\right]}{V_{fg}} \quad [-] \tag{10}$$

Volumetric proportion of nitrogen in flue gases

$$X_{N2} = \frac{\left(0.8 \cdot \frac{N^{daf}}{100} + 0.79\lambda \cdot V_{air}\right) \cdot \left[1 - \frac{A}{100} \cdot \left(1 - \frac{W}{100}\right) - \frac{W}{100}\right]}{V_{fg}} \quad [-] \tag{11}$$

Volumetric proportion of oxygen in flue gases

$$X_{O2} = \frac{0.21 \cdot V_{air} \cdot (\lambda - 1) \cdot \left[1 - \frac{A}{100} \cdot \left(1 - \frac{W}{100}\right) - \frac{W}{100}\right]}{V_{fg}} \quad [-] \tag{12}$$

Volumetric proportion of water vapor in flue gases

$$X_{H2O} = \frac{11.2 \cdot \frac{H^{daf}}{100} \cdot \left[1 - \frac{A}{100} \cdot \left(1 - \frac{W}{100}\right) - \frac{W}{100}\right] + 1.24\frac{W}{100}}{V_{fg}} \quad [-] \tag{13}$$

The temperature of flue gases ($t_{fg}$) emitted to the atmosphere by fluid or steam boilers is dependent on the construction of the boiler heat exchanger and the temperature of the heated water, thermal oil, or water vapor produced. The temperature of the flue gases leaving the boiler ranges from $t_{fg}$ = 120–200 °C, according to the prestigious producers of firewood boilers, such as Herz GmbH, Vincke Energietechniek n.v., TTS Group Třebíč, Vissmann, and Justsen Energiteknik A/S.

The heating value of firewood is affected by the moisture content, as it has been described by many authors [1,6,12], using the following equation:

$$Q_n = 18840 - 21351 \cdot \frac{W}{100} \qquad \left[\text{kJ} \cdot \text{kg}^{-1}\right] \qquad (14)$$

The thermal efficiency of a boiler with a controlled combustion process of firewood, in accordance with the environmental benefits and ecological criteria, BAT (the best available technologies), as it is mentioned in the literature [16,17,27], is dependent especially on the chimney heat loss. The dependence of the thermal efficiency of a boiler when the nominal thermal output performance ranges from $P_{nom}$ = 1–5 MW on the temperature of the emitted flue gases from a boiler into the atmosphere at $t_{fg}$ = 120–200 °C, and the moisture content of firewood of W = 10%–60%, is described using the following equation:

$$\eta_k = \left[-0.001 \cdot (W)^2 - 0.0019 \cdot W + 91.52\right] - \left[(0.001 \cdot W + 0.058) \cdot \left(t_{fg} - 120\right)\right] \qquad [-] \qquad (15)$$

## 3. Dependence of Producing Flue Gases and Atmospheric Thermal Load by the Heat of Emitted Flue Gases on the Moisture Content of Combusted Firewood

The volume of flue gases of $V_{fg\text{-}1GJ}$ = 656.43 $\text{m}_n^3$ is produced and delivered to the atmosphere as a result of the combustion process of dried firewood, with the following chemical composition of flammable substances, namely, $C^{daf}$ = 50.0% ± 1.0%, $H^{daf}$ = 6.0% ± 0.1%, and $O^{daf}$ = 44.0 ± 3.0, and an ash content in wood of A = 1.0% with an excess of combustion air $\lambda$ = 2.1 necessary to produce 1 GJ of heat. The effect of the firewood moisture content ranging from W = 10%–60% on the energetic properties of the combusted wood, material, and technical conditions of heat production, and the atmospheric thermal load by flue gases with the temperature $t_{fg}$ = 120 °C, is mentioned in Table 2.

**Table 2.** The effect of firewood moisture content on the energy efficiency of a boiler, fuel consumption, and the production of flue gases when 1 GJ of heat is produced.

| Fuel | Moisture Content | Heating Value | Specific Volume of Flue Gases from 1 kg of Wood | Thermal Efficiency of a Boiler | Consumption of Firewood to Produce 1 GJ of Heat | Production of Emitted Flue Gases when 1 GJ of Heat is Produced | Atmospheric Thermal Load by Flue Gases |
|---|---|---|---|---|---|---|---|
| | $W^r$ | $Q_n$ | $V_{fg}$ | $\eta_k$ | $m_{p\text{-}1GJ}$ | $V_{fg\text{-}1GJ}$ | $Q_{fg}$ |
| | [%] | [kJ·kg$^{-1}$] | [$\text{m}_n^3$] | [-] | [kg] | [$\text{m}_n^3$] | [MJ·GJ$^{-1}$] |
| Fuel wood | 10 | 16,159 | 9.34 | 0.883 | 73.38 | 704.02 | 96.2 |
| | 20 | 14,075 | 8.44 | 0.878 | 87.49 | 738.47 | 101.3 |
| | 30 | 11,992 | 7.54 | 0.871 | 107.05 | 807.12 | 110.1 |
| | 40 | 9908 | 6.64 | 0.861 | 129.06 | 856.98 | 122.9 |
| | 50 | 7825 | 5.74 | 0.845 | 169.27 | 971.58 | 143.2 |
| | 60 | 5742 | 4.8 | 0.819 | 245.03 | 1188.87 | 179.8 |

Correlation between the volume of emitted flue gases resulting from the production of the heat of 1 GJ and the moisture content of combusted wood is shown in the graph in Figure 1.

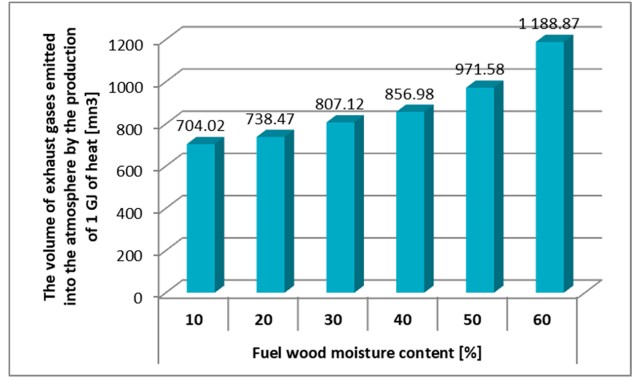

**Figure 1.** The correlation between the production of flue gases emitted in the combustion process of the firewood with the combustion air excess $\lambda$ = 2.1 when 1 GJ of heat is produced, and the moisture content of firewood.

Following the findings, we can state that in order to produce 1GJ of heat in the boiler furnace, a quantity of m = 73.5 kg of wood with a moisture content of W = 10% is burnt, and a volume of $V_{fg\text{-}1GJ}$ = 704.02 $m_n^3$ of flue gas is produced. Because of the lower heat value of the combusted wet wood with a moisture content of W = 60%, and a decrease in the energy efficiency of a boiler by o $\Delta\eta_k$ = 6.4%, 3.3 times more fuel is consumed, and a volume of $V_{fg\text{-}1GJ}$ = 1188.88 $m_n^3$ of flue gas is delivered to the atmosphere in order to produce the same amount of heat. Therefore, an increase in the flue gases of $\Delta V_{fg\text{-}1GJ}$ = 486 $m_n^3$ in comparison to the combustion of dried wood is observed. An increase in flue gas production resulting from the combustion process of wetter wood is caused by a higher volume of water vapor in the flue gases from the evaporated water occurring in combusted wood, as well as by the heated nitrogen and unoxidized oxygen in the combustion air delivered to the boiler furnace in order to produce heat consumed in the drying process of firewood. The dependence of the atmospheric thermal load from the heat of the flue gases emitted to the atmosphere with the temperature of $t_{fg}$ = 120 °C on the moisture content of combusted wood is illustrated in Figure 2.

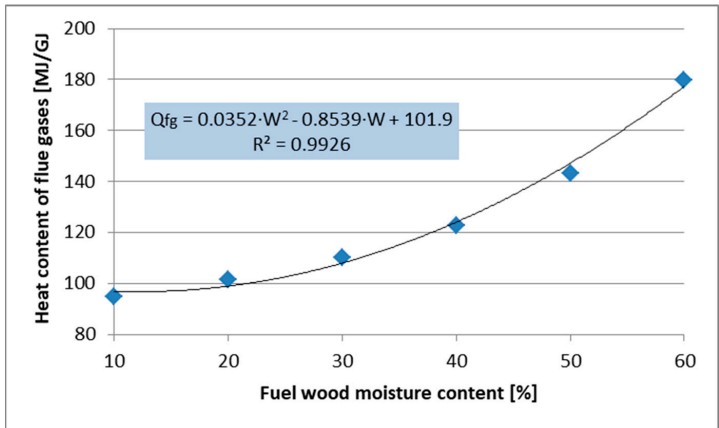

**Figure 2.** The dependence of the atmospheric thermal load by flue gases with the temperature of $t_{fg}$ = 120 °C on the moisture content of firewood.

A higher production of emitted flue gases from a boiler in the combustion process of firewood with a higher moisture content negatively affects the higher heat transfer by emitted flue gases into the atmosphere. This is confirmed by the gathered data, comparing the values of the atmospheric thermal load of flue gases with temperatures of $t_{fg}$ = 120 °C in the combustion process of dried wood with a moisture content of W = 10%, to flue gases in the combustion process of wet wood with a moisture content of W = 60%. The atmospheric thermal load created by flue gases in the combustion process of dried wood is $Q_{fg}$ = 96.2 MJ·GJ$^{-1}$. On the other hand, the atmospheric thermal load created by flue gases in the combustion process of wet wood is $Q_{fg}$ = 179.8 MJ·GJ$^{-1}$. This indicates an increase in flue gases of $\Delta Q_{fg}$ = 83.6 MJ·GJ$^{-1}$, as well as more heat delivered to the atmosphere.

Figure 3 illustrates the effect of the temperature of flue gases leaving a boiler, with temperatures ranging from $t_{fg}$ = 120–200 °C, on the atmospheric thermal load, due to the combustion process of firewood with a moisture content of $W^r$ = 10%–60% in the boiler furnace in the combustion process of firewood, with an excess of combustion air $\lambda$ = 2.1 and average temperature of air delivered to the furnace $t_{vz}$ = 10 °C.

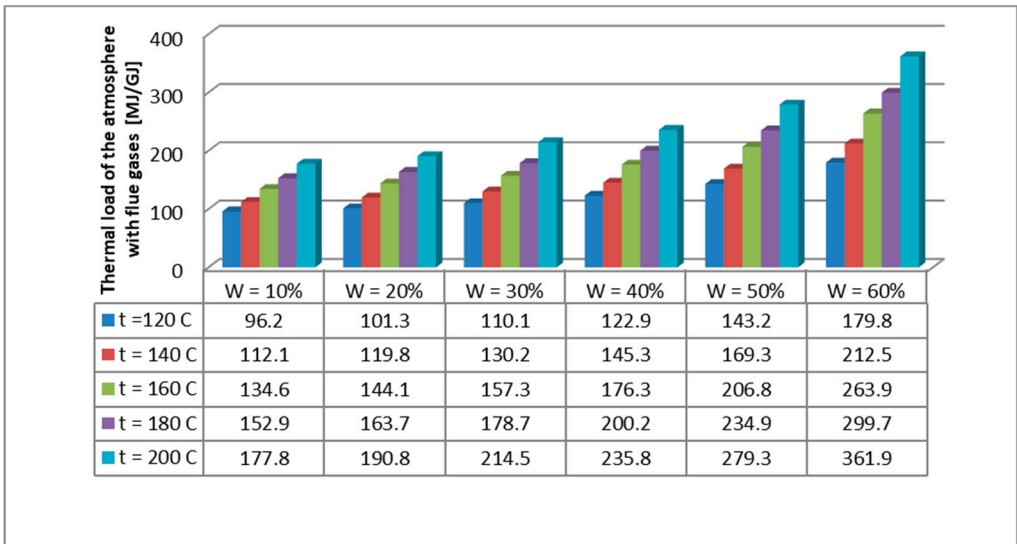

**Figure 3.** The correlation between the atmospheric thermal load created by flue gases with the temperature of $t_{fg}$ = 120–200 °C and combusted wood with moisture content of W = 10%–60%.

Because of the higher temperatures of the emitted flue gases from a boiler, the atmospheric thermal load and heating of the atmosphere is higher. While the atmospheric thermal load created by the flue gases in the combustion process of dried wood and the temperature of flue gases $t_{fg}$ = 120 °C is $Q_{fg}$ = 96.2 MJ·GJ$^{-1}$, the value of the atmospheric thermal load created by the flue gases in the combustion process of wet wood with a moisture content of W = 60 %, and the temperature of the flue gases $t_{fg}$ = 200 °C is up to $Q_{fg}$ = 3619 MJ·GJ$^{-1}$. In comparison to the atmospheric thermal load by the flue gases in the combustion process of dried wood, an increase of 3.7 times can be seen.

When comparing the average atmospheric thermal load created by the heat of flue gases when the moisture content of combusted wood increases, we can state that the emitted heat in the flue gases with temperatures of $t_{fg}$ = 120 °C, and with an increase in the moisture content of firewood by 1%, increases by $\Delta Q_{fg}$ = 1.7 MJ. When the temperature of the emitted flue gases is $t_{fg}$ = 200 °C, then the increase observed is $\Delta Q_{fg}$ = 3.68 MJ. Following the presented data that describes the atmospheric thermal load created by the flue gases emitted from a boiler, depending on the moisture content of the combusted wood and the temperature of the flue gases, the moisture content of the combusted wood of W = 10–60% and the temperature of the emitted flue gases $t_{fg}$ = 120–200 °C was used to derive the functional dependence in the form of a 3D graph Figure 4, using the program STATISTICA, for boundary conditions, and Equation (16).

$$Q_{fg} = 271.69 - 3.79 \cdot W + 2,28 \cdot t_{fg} + 0.04 \cdot W^2 + 0.02 \cdot W \cdot t_{fg} + 0.01 \cdot t_{fg}^2 \qquad \left[\text{MJ} \cdot \text{GJ}^{-1}\right] \qquad (16)$$

In order to compare the production of the flue gases and the atmospheric thermal load created by the flue gases resulting from the production of the heat of 1 GJ in the combustion process of firewood to other fuels, Table 3 shows the production of flue gases in a natural gas boiler with a thermal efficiency of $\eta_K$ = 95%, and the atmospheric thermal load created by flue gases with temperatures of $t_{sp}$ = 110 °C emitted from the boiler to the atmosphere.

The volume of flue gases emitted to the atmosphere as a result of the production of the heat of 1 GJ from natural gas is 2.4 times lower than the volume of flue gases produced and delivered to the atmosphere when 1 GJ of heat was produced from dried firewood with a moisture content of W = 10%. The volume value was four times lower than the value from the production of heat from wet wood with a moisture content of W = 60%. The values of the atmospheric thermal load created by the flue gases emitted in the production of 1 GJ of heat from natural gas with a temperature of $t_{fg}$ = 110 °C, or of dried and wet firewood with a temperature of $t_{fg}$ = 120 °C, are mentioned in the bar chart in Figure 5.

The atmospheric thermal load created by the flue gases resulting from the combustion process of dried wood, with the temperature of flue gases of $t_{fg}$ = 120 °C is 1.8 times higher, and the atmospheric thermal load created by the flue gases resulting from the combustion process of wood with a moisture content of W = 60%, and the temperature of flue gases of $t_{fg}$ = 200 °C is seven times higher than in the combustion of natural gases.

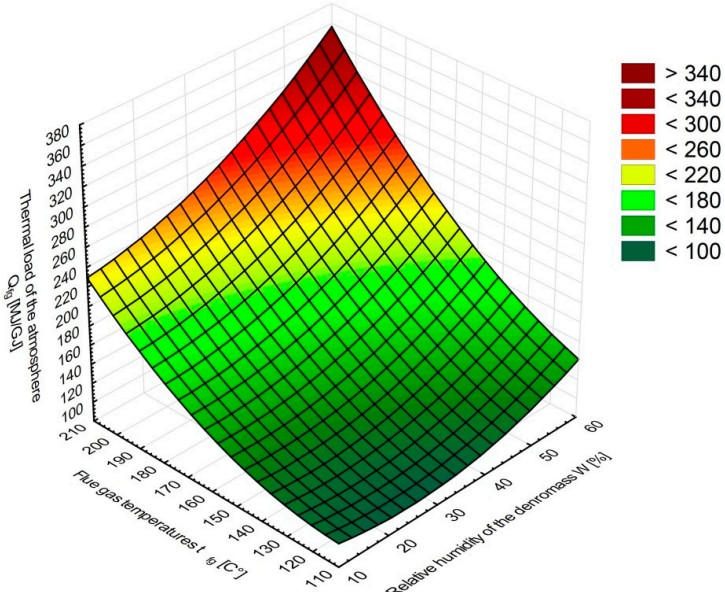

**Figure 4.** 3D graph of the dependence of atmospheric thermal load ($Q_{fg}$) on the moisture content of the combusted firewood with a moisture content of W = 10%–60%, and the temperature of emitted flue gases from a boiler, $t_{fg}$ = 120–200 °C.

**Table 3.** The production of flue gases and the atmospheric thermal load created by flue gases from a natural gas boiler.

| Fuel | Heating Value | Production of Emitted Flue Gases When 1 GJ of Heat is Produced | Thermal Efficiency of a Boiler | Atmospheric Thermal Load |
|---|---|---|---|---|
| | [MJ·m$^{-3}$] | [m$_n^3$] | [-] | [MJ·GJ$^{-1}$] |
| Natural gas | 33.9 | 297 | 0.95 | 51.6 |

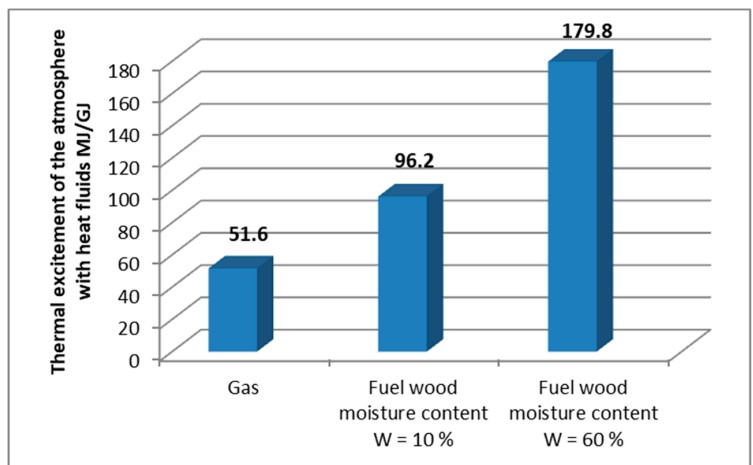

**Figure 5.** The atmospheric thermal load created by the heat of emitted flue gases from a natural gas boiler and a boiler fed with firewood with moisture contents of W = 10% and W = 60%.

The above arguments on the influence of wood moisture on the warming of the atmosphere by the flue gases emitted from the boiler illustrate the fact that the combustion of moist wood reduces the thermal efficiency of the boiler [16,27] and increases the production of the emission [20,23]. This is the reason for introducing the economically efficient pre-drying and seasoning of fuel wood. Such technologies, as mentioned in the literature [6,27], include the technology of the transpiration drying of the branches and top of trees before the production of wood chips, as well as the natural drying of stored firewood on the covered storehouses.

## 4. Conclusions

Following the results of the analysis of the effect of the moisture content of firewood on the atmospheric thermal load created by the heat of flue gases emitted from a firewood boiler, the following statements can be made:

(1) in the production of 1 GJ of heat from dried wood with a moisture content of W = 10%, a volume of flue gases of $V_{fg\text{-}1GJ}$ = 96.2–177.8 $m_n^3$ with a temperature of $t_{fg}$ = 120–200 °C is delivered to the atmosphere. In the combustion process of wet wood with a moisture content of W = 60%, a volume of flue gases of $V_{fg\text{-}1GJ}$ = 179.8–361.9 $m_n^3$ is emitted to the atmosphere.

(2) An increase of 1% in the moisture content of firewood results in an increase in the atmospheric thermal load created by the heat of flue gases with a temperature of $t_{fg}$ = 120 °C of $\Delta Q_{fg}$ = 1.7 MJ, on average. When the temperature of the flue gases is $t_{fg}$ = 200 °C, it increases by $\Delta Q_{fg}$ = 3.68 MJ.

(3) Comparing the values of the atmospheric thermal load created by the flue gases resulting from the combustion process of firewood to the thermal load caused by the combustion process of natural gas, we can state that the atmospheric thermal load caused by the combustion of firewood ranges from 1.8–7 times higher.

(4) The heat of the water vapor from the evaporated water of the combusted wood, as well as the heat of the heated nitrogen and unoxidized oxygen in the combustion air delivered to the furnace of a boiler to dry firewood, are the reasons for the increasing volume of the flue gases. This, therefore, causes the increasing atmospheric thermal load created by the heat of the flue gases, resulting from the combustion of wood with higher moisture content.

(5) Because of the effect of the moisture content of the firewood on the atmospheric thermal load created by emitted the flue gases from firewood, as well as the fact that, because of the moisture content, the energy efficiency of the boiler and the efficiency of the heat production decreases and the emission production increases. This justifies the use of economically-efficient forms of pre-drying and seasoning of firewood, such as the technology of transpiration used to dry branches and tree tops before the production of wood chips, or the natural pre-drying of stored firewood logs in a sheltered position.

**Author Contributions:** Conceptualization, A.B. and L.D.; methodology, A.B. and L.D.; software, A.B.; validation, A.B; formal analysis, A.B. and L.D.; investigation, A.B. and L.D.; resources, A.B. and L.D.; data curation, A.B. and L.D.; writing (original draft preparation), A.B. and L.D.; writing (review and editing), A.B. and L.D.; supervision, L.D.; project administration, L.D.

**Funding:** This research was funded by project KEGA-SR no 003TU Z-4/2018, as a result of the author's work, as well as from the significant assistance of the Cultural and Educational Grant Agency of the Ministry of Education, Science, Research, and Sport of the Slovak Republic.

**Conflicts of Interest:** The funders had no role in the design of the study; in the collection, analyses, or interpretation of data; in the writing of the manuscript; or in the decision to publish the results.

## Nomenclature

| | |
|---|---|
| A | ash content in dried firewood, % |
| $c_{fg}$ | specific heat capacity 1 $mn^{-3}$ of flue gases when the pressure is constant, $kJ \cdot m_n^{-3} K^{-1}$ |
| $c_{pCO2}$ | specific heat capacity of carbon dioxide, $kJ/m^3 \cdot K$ |
| $c_{pH2O}$ | specific heat capacity of water vapor, $kJ/m^3 \cdot K$ |

| | |
|---|---|
| $c_{pO2}$ | specific heat capacity of oxygen, kJ/m$^3$·K |
| $c_{pN2}$ | specific heat capacity of nitrogen, kJ/m$^3$·K |
| $C^{daf}$ | amount of carbon in flammable substance of firewood, % |
| $H^{daf}$ | amount of hydrogen in flammable substance of firewood, % |
| $N^{daf}$ | amount of nitrogen in flammable substance of firewood, % |
| $O^{daf}$ | amount of oxygen in flammable substance of firewood, % |
| $m$ | quantity of wood to produce 1 GJ of heat, kg |
| $t_{air}$ | temperature of combustion air, °C |
| $t_{fg}$ | temperature of flue gases, °C |
| $t_{fg-e}$ | temperature of flue gases cooled to the temperature of combustion air, °C |
| $V_{air}$ | stechiometric combustion air volume, m$^3$/kg |
| $V_{fg}$ | specific volume of humid flue gases produced in the combustion of 1kg of firewood, m$^3$/kg |
| $V_{fg-1MJ}$ | flue gases produced in the firewood combustion to produce the heat of 1GJ, mn$^3$ |
| W | moisture content of firewood, % |
| $Q_n$ | heating value, kJ/kg, or MJ/m$^3$ |
| $X_{CO2}$ | volumetric proportion of carbon dioxide in flue gases, m$^3$/m$^3$ |
| $X_{CO}$ | mass concentration of carbon monoxide (CO) in dry flue gases, kg/m$^3$ |
| $X_{H2O}$ | volumetric proportion of water vapor in flue gases, m$^3$/m$^3$ |
| $X_{O2}$ | volumetric proportion of oxygen in flue gases, m$^3$/m$^3$ |
| $X_{N2}$ | volumetric proportion of nitrogen in flue gases, m$^3$/m$^3$ |

**Greek symbols**

| | |
|---|---|
| $\lambda$ | coefficient of the excess of combustion air, m$^3$/m$^3$ |
| $\eta_K$ | energy efficiency of a boiler, % |

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
