# Peer review of "The Effect of Firewood Moisture Content on the Atmospheric Thermal Load by Flue Gases Emitted by a Boiler"

_sustainability, doi:10.3390/su11010284_

Reviewer 1 Report

see attached

Author Response

Response to Reviewer 1 Comments

Point 1:

How about wood species that could affect the result.

How much and significant if compared?

Response 1: Please provide your response for Point 1

Wood material independently of the tree species is still chemical composition. It is composed of elements: C = 50.0 ± 1.0%, H = 6.0 ± 0.1%, O = 44.0 ± 3.0, The proportion of auxiliary substances such as resin in pine wood or spruce is negligible and does not exceed the tolerance limits of carbon C and hydrogen H.

Point 2:

Need to define the thermal load.

Thermal load discharged from boiler that include all the thermal energy produced by the firewood or the energy left after some energy used by the boiler. 

Also different type of boiler will have difference efficiency on the energy usage. 

Response 2: Please provide your response for Point 2.

Linees 56, 57 and 58 will be replaced by:

The heat present in the exhaust gases from the boiler to the atmosphere is the thermal load of the atmosphere. The heat of emitted flue gas related to the production of 1 GJ of heat is mathematically described by equation (1):

and:

Different types of boilers are projected at a different temperature of the gases emitted into the atmosphere. In the heat load models of the atmosphere, this is reflected by the temperature of the flue gas.

Point 3:

reference

Response 3: Please provide your response for Point 3.

Added references:

[1, 6, 27, 28 ]

[27] Sergovskij P.S., Rasev A.I.: Gidrotermičeskaja obrabotka i konservirovanije drevesiny, Moskva: Lesnaja promyšlennost, 1987, 350 p.

[28] Dzurenda, L., Banski, A.: Výroba tepla a energie z dendromasy. Zvolen: TU vo Zvolene, 2016, 273 p.

Point 4:

explain the term in all formula

Response 4: Please provide your response for Point 4.

The terms are for all equations listed in the NOMENGLATURE section

Point 5:

confused.  is it the ambient air temperature???

Response 5: Please provide your response for Point 5.

The flue gas temperature tfg-e is equal to the temperature of the combustion air tair. In equation (1), the heat content of the flue gases emitted into the atmosphere is balanced by the temperature difference of the flue gases emitted by the flue gases into the atmosphere and the temperature of the flue gases tfg-e in their theoretical cooling to the temperature of the incoming atmospheric air. If the flue gas is cooled to the intake air temperature, the thermal load of the atmosphere would be 0 (Qfg = 0).

Reviewer 2 Report

The high initial moisture content in biomass is always a challenge, which limits the uses of biomass as fuel in a certain region where is close to forest with a consideration of long-distance transportation cost. Direct combustion of raw biomass consumes a large portion of heat energy to convert the free water in wood to steam (i.e., sensible and latent heat in steam) and increase the sensible heat in other gases, lowering the energy efficiency and increasing the thermal load of atmosphere. This study calculated the thermal energy released to the atmosphere by burning firewood with moisture content varying from 10% to 60% and discussed the effects of temperature of flue gases on the thermal energy released. A regression model with moisture content and temperature of flue gases as variables was developed, which can be used to calculate the atmospheric thermal load generated by burning hardwood biomass. Overall, this is a high-quality manuscript. The deduction of the calculation is concise and accurate, and the discussion is sufficient. The future work would be how to pre-dry the biomass by means of an economic approach.

The authors can add a description of the size of the firewood or a picture to show the readers what type of firewood was studied.

Author Response

Thank you for reviewing the article. Fuel wood is in the form of an energy chips. Image not required.